# A Low-Complexity FMCW Surveillance Radar Algorithm Using Two Random Beat Signals

**DOI:** 10.3390/s19030608

**Published:** 2019-01-31

**Authors:** Bong-seok Kim, Youngseok Jin, Sangdong Kim, Jonghun Lee

**Affiliations:** Advanced Radar Technology Laboratory, DGIST, Daegu 42988, Korea; remnant@dgist.ac.kr (B.-s.K.); ysjin@dgist.ac.kr (Y.J.); kimsd728@dgist.ac.kr (S.K.)

**Keywords:** FMCW radar, surveillance, low complexity, Doppler

## Abstract

This paper proposes a low-complexity frequency-modulated continuous wave (FMCW) surveillance radar algorithm using random dual chirps in order to overcome the blind-speed problem and reduce the computational complexity. In surveillance radar algorithm, the most widely used moving target indicator (MTI) algorithm is proposed to effectively remove clutter. However, the MTI algorithm has a so-called ‘blind-speed problem’ that cannot detect a target of a specific velocity. In this paper, we try to solve the blind-speed problem of MTI algorithm by randomly selecting two beat signals selected for MTI for each frame. To further reduce the redundant complexity, the proposed algorithm first performs one-dimensional fast Fourier transform (FFT) for range detection and performs multidimensional FFT only when it is determined that a target exists at each frame. The simulation results show that despite low complexity, the proposed algorithm detects moving targets well by avoiding the problem of blind speed. Furthermore, the effectiveness of the proposed algorithm was verified by performing an experiment using the FMCW radar system in a real environment.

## 1. Introduction

Recently, several studies on radar sensors have been reported [1,2,3,4]. Radar sensors allow safe detection of targets because radar sensors are less sensitive to increment conditions such as heavy rain, snow, and fog compared to other sensors such as camera and LiDAR [5]. Due to these advantages, radar sensors have been employed for several applications. For example, radar sensors have been used for automotive applications such as adaptive cruise control, avoidance of collision and parking aid [6]. In addition, radar sensors are employed in not only military applications such as detection of enemy tank and airborne but also surveillance applications [1,2,3].

As one of the most promising among the various radar techniques, frequency-modulated continuous wave (FMCW) radar systems have been studied [6,7,8,9]. In FMCW radar systems, by multiplying the received signal by the transmitted signal, linear combination of sinusoid signals with low frequencies, so-called ‘beat signal’ can be directly obtained. Hence, FMCW radar systems can reduce the cost of hardware and architecture because the obtained beat signal can be digitized directly compared to pulsed radar.

In earlier studies [5,10,11], FMCW radar algorithms for surveillance applications were designed and addressed. In one work [5,10], the authors investigated the implementation of a 24 GHz FMCW radar for surveillance, and detected the range, velocity, and angle of targets. Meanwhile, in another work [11], the authors presented a scalable architecture for acquisition and a field programmable gate array (FPGA)-based processing platform of a radar sensor with a single transmitter and multiple receivers. However, these algorithms [5,10,11] perform full-dimensional fast Fourier transforms (FFTs) when detecting to distinguish between stationary and moving targets, and thus require high complexity. The most important issue for surveillance applications is rapid identification of the presence of a moving target. Therefore, these algorithms are not suitable for low cost surveillance applications due to their high complexity.

Meanwhile, in [12,13,14], FMCW radar algorithms with low complexity have been proposed by reducing the number of FFTs compared to the conventional FMCW radar algorithms using full dimension FFT. By performing FFTs on only regions of interest (ROIs) instead of full dimension FFTs, these algorithms reduce the redundant complexity compared to the conventional FMCW radar algorithms using full dimension FFT. However, to apply these algorithms to surveillance applications, an additional algorithm is required to distinguish between stationary target and moving target.

To distinguish between stationary target and moving target, the moving target indicator (MTI) algorithms have been employed and discussed [4] and their applications include the surveillance, indoor tracking and vehicle radar systems [9]. More recently, in [8,9], MTI algorithms for FMCW radar systems have been proposed. In [8], this algorithm effectively detects the moving targets by performing FFT on only difference between two beat signals. In the case of a stationary target, the difference of two beat signals becomes zero and, thus, the FFT result includes only additive white Gaussian noise (AWGN). On the other hand, in the case of a moving target, the result of the difference between two beat signals includes information of the ranges of targets. Hence, the range of moving target is effectively detected. By employing only two beat signals, this algorithm has significantly reduced the complexity of FMCW radar systems. However, there are still drawbacks that need to be improved. First, this algorithm cannot overcome the blind-speed problem [3]; a target with a specific velocity may not be detected because two beat signals are selected regardless of the target’s velocity. Moreover, there is also redundant complexity in [8] that can be reduced because this algorithm performs a 2D FFT to detect the range and angle of a target, regardless of whether it is present.

In this paper, the proposed algorithm tries to overcome the problem of blind speed, i.e., the drawbacks of [8] while further reducing complexity. To this end, first, to solve the blind-speed problem, two beat signals are randomly chosen unlike [8]. By randomly selecting two beat signals at each frame, even if detection fails in a frame, the proposed algorithm detects the moving target in another frame. Secondly, the proposed algorithm further reduces the complexity compared to [8]. Instead of performing the 2D FFT to detect the range and angle at every frame as in [8], the proposed algorithm performs the 2D FFT only when it is determined that the target is present. Moreover, to verify the effectiveness of the proposed algorithm, we perform simulations and an experiment in a real environment. The results of simulation and experiment show that the proposed algorithm achieves better performance compared to the [8] despite further low complexity. These results imply that the proposed algorithm is one of solutions to the blind speed problem that misses a target a specific velocity.

The structure of the paper is as follows. In Section 2.2, we introduce and define the system model and the main notations used. Furthermore, we establish the FMCW and detection algorithms using a 3D FFT for the FMCW radar systems. Section 3 considers the low-complexity surveillance FMCW radar algorithm using two beat signals as proposed in Reference [8] and describes its shortcomings. In Section 4, we introduce the structure of the proposed algorithm and describe how it overcomes the problems in [8]. In Section 5, simulations are performed to evaluate performance and show the improvements of the proposed algorithm compared to Reference [8]. Section 6 provides the experimental results for various cases to verify the effectiveness of the proposed algorithm by implementation of a 24 GHz FMCW radar system. Finally, Section 7 concludes this paper.

## 2. System Model and Conventional 3D-FFT-Based Detection Algorithm in FMCW Radar Systems

### 2.1. System Model

This section describes the system model in FMCW radar systems. As shown in Figure 1, the total transmitted (TX) FMCW signal is denoted by x(t) and expressed as:(1)$$\begin{array}{c} x\left(t\right)=\sum_{i=0}^{N_{F}-1}x^{\left(i\right)}\left(t-iT_{F}\right), \end{array}$$
where $T_{F}$ is the duration of a frame as shown in Figure 1 and $N_{F}$ is the number of frames. The TX FMCW signal which is composed of *L* chirps at the *i*th frame, $x^{\left(i\right)}\left(t\right)$ is expressed as:(2)$$\begin{array}{c} x^{\left(i\right)}\left(t\right)=\sum_{l=0}^{L-1}x_{0}\left(t-lT\right), \end{array}$$
where *T* is the duration of an FMCW chirp signal $x_{0}\left(t\right)$. An FMCW chirp signal $x_{0}\left(t\right)$ is expressed as follows [15]:(3)$$\begin{array}{c} x_{0}\left(t\right)=\exp\left(j2\pi \left(f_{0}t+\frac{\mu }{2}t^{2}\right)\right)\;\;\mathrm{for}\;\;0\leq t\leq T, \end{array}$$
where $f_{0}$ is the carrier frequency and μ is the rate of change of the instantaneous frequency of a chirp signal, i.e., μ=B/T, where *B* is the bandwidth of the FMCW chirp signal.

We consider *M* far-field, non-coherent, narrow-band targets impinging on a uniform linear array (ULA) with *K* elements. The received (RX) signal of the *k*th array for the *l*th chirp at the *i*th frame is denoted by $r_{l,k}^{\left(i\right)}\left(t\right)$ and is expressed as [15]:(4)$$\begin{array}{c} r_{l,k}^{\left(i\right)}\left(t\right)=\sum_{m=1}^{M}{\tilde{a}}_{m}^{\left(i\right)}x_{0}\left(t-\tau_{m}^{\left(i\right)}\right)\exp\left(j2\pi f_{D,m}^{\left(i\right)}\left(Tl+\left(i-1\right)T_{F}\right)\right)\exp\left(j\frac{2\pi }{\lambda }d_{s}k\sin\theta_{m}^{\left(i\right)}\right)+{\tilde{w}}_{l,k}^{\left(i\right)}\left(t\right), \end{array}$$
where ${\tilde{a}}_{m}^{\left(i\right)}$ is the complex amplitude of the reflected signal of the *m*th target in the *i*th frame, $d_{s}$ is the spacing between the adjacent arrays, λ is the wavelength of the carrier frequency, and $\tau_{m}^{\left(i\right)}$, $f_{D,m}^{\left(i\right)}$, and $\theta_{m}^{\left(i\right)}$ are the round-trip time delay, Doppler frequency due to the velocity of the moving target, and the direction-of-arrival (DOA) for the *m*th target in the *i*th frame, respectively. Additionally, ${\tilde{w}}_{l,k}^{\left(i\right)}\left(t\right)$ is the AWGN signal for the *k*th array, the *l*th chirp, and the *i*th frame. By multiplying the conjugated FMCW TX signal $x_{0}^{*}\left(t\right)$ by $r_{l,k}^{\left(i\right)}\left(t\right)$ and assuming $d_{s}=\lambda /2$, the beat signal for the *l*th chirp, the *k*th array, and the *i*th frame, $y_{l,k}^{\left(i\right)}\left(t\right)$ is obtained and expressed as the product of the range, Doppler, and DOA terms as follows:(5)$$\begin{array}{ll} y_{l,k}^{\left(i\right)}\left(t\right) & =r_{l,k}^{\left(i\right)}\left(t\right)\times x_{0}^{*}\left(t\right)\\ & =\sum_{m=1}^{M}a_{m}^{\left(i\right)}\underset{\;\mathrm{range}\;\mathrm{term},≜\eta_{m}^{\left(i\right)}\left(t\right)}{\underset{︸}{\exp\left(-j2\pi f_{b,m}^{\left(i\right)}t\right)}}\underset{\mathrm{Doppler}\;\mathrm{term},\;≜v_{m}^{(i)l}}{\underset{︸}{\exp\left(j2\pi f_{D,m}^{\left(i\right)}\left(Tl+T_{F}\left(i-1\right)\right)\right)}}\underset{\mathrm{DOA}\;\mathrm{term},\;≜z_{m}^{(i)k}}{\underset{︸}{\exp\left(j\pi k\sin\theta_{m}^{\left(i\right)}\right)}}\\ & +\underset{\mathrm{noise}\;\mathrm{term},\;≜w_{l,k}^{\left(i\right)}}{\underset{︸}{{\tilde{w}}_{l,k}^{\left(i\right)}\left(t\right)x_{0}^{*}\left(t\right)}}\\ & =\sum_{m=1}^{M}a_{m}^{\left(i\right)}\eta_{m}^{\left(i\right)}\left(t\right)v_{m}^{(i)l}z_{m}^{(i)k}+w_{l,k}^{\left(i\right)}\left(t\right), \end{array}$$
where $a_{m}^{\left(i\right)}$ is defined as $a_{m}^{\left(i\right)}={\tilde{a}}_{m}^{\left(i\right)}\exp\left(-j\left(2\pi f_{0}\tau_{m}^{\left(i\right)}-\mu \tau_{m}^{(i)2}/2\right)\right)$ as in [15] and $f_{b,m}^{\left(i\right)}$ is the beat frequency due to delay, i.e., $f_{b,m}^{\left(i\right)}=\mu \tau_{m}^{\left(i\right)}$ in $\eta_{m}^{\left(i\right)}\left(t\right)$ in Figure 1.

After analog to digital conversion (ADC) of $y_{l,k}^{\left(i\right)}\left(t\right)$, the discrete time model of Equation (Equation 5) with sampling frequency $f_{s}$ is denoted by $y_{l,k}^{\left(i\right)}\left[n\right]$, i.e., $y_{l,k}^{\left(i\right)}\left[n\right]=y_{l,k}^{\left(i\right)}\left(nT_{s}\right)$ for $n=0,1,\ldots ,N_{s}-1$, where $T_{s}=1/f_{s}$ is the sampling interval, and $N_{s}=\left\lceil T/T_{s}\right\rceil$ is the number of samples where ⌈·⌉ is ceil operator. Thus, Equation (Equation 5) is rewritten as:(6)$$\begin{array}{c} y_{l,k}^{\left(i\right)}\left[n\right]=\sum_{m=1}^{M}a_{m}^{\left(i\right)}\eta_{m}^{\left(i\right)}\left[n\right]v_{m}^{(i)l}z_{m}^{(i)k}+w_{l,k}^{\left(i\right)}\left[n\right]. \end{array}$$

### 2.2. Conventional 3D-FFT-Based Detection Algorithm in FMCW Radar Systems

This section illustrates the conventional 3D-FFT-based detection algorithm in FMCW radar systems. From Equation (Equation 6), the ADC beat signal can be expressed as a sinusoidal signal in three-dimensions (3D), i.e., range in the sample domain *n*, velocity in the chirp domain *l*, and angle in the array domain *k*, as shown in Figure 2. Therefore, by estimating the frequencies of the 3D sinusoidal signals using a 3D FFT, the desired parameters can be detected. Figure 2 illustrates the structure of parameter estimation using 3D FFT in the *i*th frame. In Figure 2, $N_{R}$, $N_{C}$, and $N_{A}$ are the number of FFT points on the range, chirp, and array domains, respectively.

First, the $N_{R}$-point FFTs for range estimation on $y_{l,k}^{\left(i\right)}\left[n\right]$ are performed for 1≤l≤L and 1≤k≤K. The *q*th FFT output of $y_{l,k}^{\left(i\right)}\left[n\right]$ is denoted by $Y_{l,k}^{\left(i\right)}\left[q\right]$ and is obtained as follows [16]:(7)$$\begin{array}{c} Y_{l,k}^{\left(i\right)}\left[q\right]=\sum_{n=1}^{N_{s}}y_{l,k}^{\left(i\right)}\left[n\right]W_{N_{R}}^{\left(n-1)(q-1\right)}\;\;\mathrm{for}\;\;1\leq q\leq N_{R}, \end{array}$$
where $W_{N}$ is the *N* point discrete Fourier transform (DFT) operator, i.e., $W_{N}=\exp\left(-j2\pi /N\right)$. FFT output $Y_{l,k}^{\left(i\right)}\left[q\right]$ is called ‘range bin’ because it includes the range information of the targets as shown in Figure 2.

Secondly, in the same manner, $N_{C}$-point FFTs for Doppler estimation on the $N_{R}\times L$ FFT outputs are performed as the box indicated by the dashed lines in Figure 2. The *p*th FFT output of $Y_{l,k}^{\left(i\right)}\left[q\right]$ is denoted by ${\tilde{Y}}_{p,k}^{\left(i\right)}\left[q\right]$ and obtained for $1\leq q\leq N_{R}$ and 1≤k≤K as follows:(8)$$\begin{array}{c} {\tilde{Y}}_{p,k}^{\left(i\right)}\left[q\right]=\sum_{l=1}^{L}Y_{l,k}^{\left(i\right)}\left[q\right]W_{N_{C}}^{\left(l-1)(p-1\right)}\;\;\mathrm{for}\;\;1\leq p\leq N_{C}. \end{array}$$

From (Equation 7) and (Equation 8), *K* times 2D FFT outputs with $N_{R}\times N_{C}$ which reflect range and Doppler information are obtained. Then, these 2D FFT outputs are used as input to $N_{A}$ point FFTs for angle estimation to perform 3D FFT. Finally, by performing the peak detection and constant false-alarm rate (CFAR) algorithms, the range, velocity, and angle parameters are estimated. Meanwhile, in surveillance radar systems, it is important to detect the existence of targets as fast as possible. However, the computational complexity of 3D FFT is quite high and thus 3D FFT-based detection algorithms might not be suitable for surveillance applications.

## 3. MTI-based Low-Complexity Surveillance FMCW Radar Algorithm

### 3.1. Structure of MTI-based Low-Complexity Surveillance FMCW Radar Algorithm

This section introduces the structure of MTI-based low-complexity surveillance FMCW radar algorithm [8]. This algorithm proposes a method to efficiently detect moving targets by applying MTI algorithm to FMCW radar. In addition, the computational complexity is significantly reduced by performing an FFT on the difference between the two beat signals instead of using 3D-FFT, making it suitable for low-complexity surveillance applications. Figure 3 illustrates the structure of the low-complexity surveillance FMCW radar algorithm using two beat signals [8]. First, the $l_{1}$th and the $l_{2}$th chirp signals of the *k*th array, i.e., $y_{l_{1},k}^{\left(i\right)}\left[n\right]$ and $y_{l_{2},k}^{\left(i\right)}\left[n\right]$, are selected where $l_{u}$ is the index of the selected *u*th beat (chirp) signal for u=1,2 and $l_{u}\in \left[1,L\right]$. Then, a subtraction between these two beat signals, denoted by $d_{k}^{\left(i\right)}\left[n\right]$, is expressed as:(9)$$\begin{array}{ll} d_{k}^{\left(i\right)}\left[n\right] & =y_{l_{1},k}^{\left(i\right)}\left[n\right]-y_{l_{2},k}^{\left(i\right)}\left[n\right]\\ & =\sum_{m=1}^{M}a_{m}^{\left(i\right)}\eta_{m}^{\left(i\right)}\left[n\right]z_{m}^{(i)k}\left(v_{m}^{\left(i\right)l_{1}}-v_{m}^{\left(i\right)l_{2}}\right)+w_{l_{1},k}^{\left(i\right)}\left[n\right]-w_{l_{2},k}^{\left(i\right)}\left[n\right]. \end{array}$$

From Equation (Equation 9), while there is a change in the amplitude, it can be seen that $d_{k}^{\left(i\right)}\left[n\right]$ not only contains the range term $\eta_{m}^{\left(i\right)}\left[n\right]$ but also the angle term $z_{m}^{(i)k}$ of the target. In the case of a stationary target or clutter, the Doppler effect does not occur, i.e., $f_{D,m}^{\left(i\right)}=0$ in Equation (Equation 5) and thus, both of $v_{m}^{\left(i\right)l_{1}}$ and $v_{m}^{\left(i\right)l_{2}}$ become zero. Hence, ignoring the noise term, $y_{l_{1},k}^{\left(i\right)}\left[n\right]$ and $y_{l_{2},k}^{\left(i\right)}\left[n\right]$ are the same regardless of the other parameters such as $a_{m}^{\left(i\right)}$, $\eta_{m}^{\left(i\right)}\left[n\right]$, and $z_{m}^{(i)k}$ and thus, the difference of two beat signals $d_{k}^{\left(i\right)}\left[n\right]$ becomes also zero. Based on this property, as shown in Figure 3, the algorithm can effectively estimate the range and angle of the moving target by performing a 2D FFT on $d_{k}^{\left(i\right)}\left[n\right]$, although the algorithm uses only two beat signals where $D_{k}^{\left(i\right)}\left[q\right]$ is the *q*th FFT output of the *k*th array at the *i*th frame.

Figure 4 shows the detection results of the algorithm with a signal-to-noise ratio (SNR) = 20 dB, $N_{R}$ = 256, and $N_{A}$ = 64. The number of targets is set to 2. Their ranges are 35 and 70 m and their velocities are 5 m/s and 10 m/s, respectively. Their angles are set to $-17^{{}^{\circ}}$ and $15^{{}^{\circ}}$, respectively. Figure 4a shows the snapshot of the results of detection by 2D FFT. For easy understanding, Figure 4b,c shows the snapshot of the range and angle detection results by 1D FFT. From Figure 4, it is shown that this algorithm nearly accurately estimates the ranges and angles of the two targets. The results on the average effectiveness of this algorithm and the proposed algorithm under random condition will be shown in Section 5.

### 3.2. Drawbacks of MTI-Based Low-Complexity Surveillance FMCW Radar Algorithm

This section addresses the drawbacks of MTI-based low-complexity surveillance FMCW radar algorithm. First, this algorithm does not properly work under conditions where $d_{k}^{\left(i\right)}\left[n\right]$ becomes close to zero even if $f_{D,m}^{\left(i\right)}\neq 0$, that is, the so-called ‘blind-speed problem’ [3]. Secondly, there is also redundant computational complexity that can be reduced in this algorithm. This algorithm performs a 2D FFT to detect the range and angle of a target, regardless of whether it is present.

To effectively illustrate blind-speed problem, Figure 5 shows an example of the real part of $d_{k}^{\left(i\right)}\left[n\right]$ for a slow-moving target. For simplicity, the noise component is not included. The center frequency $f_{0}$ was set to 24 GHz, the duration *T* was set to 78 μs, and the target’s velocity $v_{r,m}^{\left(i\right)}$ was set to 5 km/h where $v_{r,m}^{\left(i\right)}=f_{D,m}^{\left(i\right)}\lambda /2$. Figure 5a shows the real part of the Doppler term $v_{m}^{(i)l}$. Since $v_{m}^{(i)l}$ is periodic signal, its amplitude is repeated with a period of about 55. If the two indices are chosen near an integer multiple of 55, the difference between the two beat signals $d_{k}^{\left(i\right)}\left[n\right]$ might be close to zero. Figure 5b shows the max value of the real part of $d_{k}^{\left(i\right)}\left[n\right]$ according to the difference between two indices $l_{▵}^{\left(i\right)}$, i.e., $\;l_{▵}^{\left(i\right)}=\left|l_{1}-l_{2}\right|$. As expected before, in the case of $l_{▵}^{\left(i\right)}=55$, $d_{k}^{\left(i\right)}\left[n\right]$ becomes almost zero. Figure 5c shows the real part of $d_{k}^{\left(i\right)}\left[n\right]$ according to sample index *n* with several cases of $l_{▵}^{\left(i\right)}$. From Figure 5c, it can be seen that the difference of amplitude of $d_{k}^{\left(i\right)}\left[n\right]$ according to $l_{▵}^{\left(i\right)}$ is significant. If $l_{▵}^{\left(i\right)}$ is set to 55, the amplitude of $d_{k}^{\left(i\right)}\left[n\right]$ significantly decreases.

To observe the case of a fast-moving target, Figure 6 shows an example of the real part of $d_{k}^{\left(i\right)}\left[n\right]$ with $v_{r,m}^{\left(i\right)}=60$ km/h. The tendency of Figure 6 is generally similar to the result of Figure 5. However, the period of $d_{k}^{\left(i\right)}\left[n\right]$ in Figure 6 decreases due to the high velocity of the target compared to the case of a slow-moving target. As shown in Figure 6c, the amplitude of $d_{k}^{\left(i\right)}\left[n\right]$ according to $l_{▵}^{\left(i\right)}$ is severely changed. Figure 5 and Figure 6 imply that the velocity of the target must be reflected in setting $l_{▵}^{\left(i\right)}$. In general, however, not only the velocity of the target is unknown but the velocity of each target is independent. Therefore, there is a risk of missing targets with a certain velocity in this algorithm because this algorithm selects two beat signals without considering the velocity of the target.

## 4. Proposed Low-Complexity FMCW Radar Algorithm for Surveillance Applications

This section addresses the proposed low-complexity FMCW radar algorithm for surveillance applications to overcome the drawbacks of [8]. For convenience, Reference [8] is called ‘the previous algorithm’ in this paper. Figure 7 shows the structure of the proposed algorithm. The proposed algorithm’s main contributions compared to the previous algorithm are as follows. First, the proposed algorithm solves the problem that the previous algorithm does not detect targets at a specific velocity. The proposed algorithm tries to decrease the probability of missing a target at a slow or fast velocity by randomly selecting the index every frame. Secondly, instead of performing a 2D FFT to detect the range and angle in every frame as in [8], the presence or absence of the target is first determined through range detection using 1D FFT; when it is determined that the target exists, then the 2D FFT is performed to detect both the range and angle of the target.

The proposed algorithm works in two modes: a case where the target is single and its velocity is roughly known, and a case where target is not single or the target’s velocity is unknown. Assuming that we know the approximate velocity of target, the period $T_{\mathrm{chirp}}$ in the chirp domain is calculated as:(10)$$\begin{array}{c} T_{\mathrm{chirp}}^{\left(i\right)}=\frac{1}{f_{D,m}^{\left(i\right)}}=\frac{\lambda }{2v_{r,m}^{\left(i\right)}}, \end{array}$$
where $f_{D,m}^{\left(i\right)}$ is Doppler frequency due to the velocity of the *m*th target $v_{r,m}^{\left(i\right)}$. In Equation (Equation 2), *T* is the duration of FMCW chirp signal and, thus, *T* becomes a sample interval in the chirp domain. Hence, the number of indices $N_{\mathrm{period}}^{\left(i\right)}$, corresponding to one period in the chirp domain is calculated as:(11)$$\begin{array}{c} N_{\mathrm{period}}^{\left(i\right)}=\left\lceil \frac{T_{\mathrm{chirp}}^{\left(i\right)}}{T}\right\rceil , \end{array}$$
where ⌈·⌉ is the ceil operator. In the first mode, therefore, $l_{▵}^{\left(i\right)}$ in the proposed algorithm is randomly selected with uniform distribution from within the following region $\left[1,\;N_{\mathrm{period}}^{\left(i\right)}\right]$: Meanwhile, in the case of a multiple-target condition or an unknown velocity of the target, $N_{\mathrm{period}}^{\left(i\right)}$ cannot be determined. In this case, therefore, the proposed algorithm randomly selects the difference of two indices $l_{▵}^{\left(i\right)}$ with uniform distribution within the entire region, i.e., $l_{▵}^{\left(i\right)}\;\;\in \;\left[1,L\right]$. Consequently, the region of $l_{▵}^{\left(i\right)}$ of the proposed algorithm is as:(12)$$\begin{array}{c} l_{▵}^{\left(i\right)}\in \left\{\begin{array}{cc} \left[1,\;\;\;\;N_{\mathrm{period}}^{\left(i\right)}\right], & \mathrm{if}\;\;\mathrm{single}\;\;\mathrm{target}\;\mathrm{and}\;\;\mathrm{known}\;\;\mathrm{velocity}\\ \left[1,\;\;\;\;L\right], & \mathrm{elsewhere} \end{array}\right.. \end{array}$$

By randomly assigning two beat signals in the total *L* chirps, the proposed algorithm tries to avoid the blind-speed problem. Even if $d_{k}^{\left[i\right]}\left[n\right]$ unfortunately becomes zero because of chosen $l_{▵}^{\left(i\right)}$, in the next frame, $l_{▵}^{\left(i\right)}$ will be newly changed, i.e., $l_{▵}^{\left(i+1\right)}\neq l_{▵}^{\left(i\right)}$. Therefore, the proposed algorithm can avoid the problem of continuously missing a target of a specific velocity.

Moreover, as shown in Figure 7, the proposed algorithm further reduces the computational complexity compared to [8] by performing angle detection only when it is estimated that a target exists, rather than detecting the angle in each frame. Since the target is not always present, the proposed algorithm reduces unnecessary processing while the target is not present.

Figure 8 shows a snapshot of the range detection results of the previous and proposed algorithms for a target with velocity $v_{r,m}^{\left(i\right)}=60$ km/h with SNR = 5 dB and $N_{R}=256$. In the previous algorithm, the two indices were set to $l_{1}=1$ and $l_{2}=25$. In Figure 8a, the detection result of the previous algorithm seems to be a noisy signal due to the blind-speed problem. On the other hand, the proposed algorithm clearly detects the target located at 40 m compared to the result of the previous algorithm.

Figure 9 shows a snapshot of range detection results of the previous and proposed algorithms with two targets. These calculations are performed with target velocities $v_{r,1}^{\left(i\right)}=3$ km/h and $v_{r,2}^{\left(i\right)}=10$ km/h, SNR = 5 dB, and $N_{R}$ = 256. The two indices in the previous algorithm were set to $l_{1}=1$ and $l_{2}=54$. In Figure 9a, the previous algorithm detects only the target located at 70 m, but misses the target located at 35 m. On the other hand, in Figure 9b, the proposed algorithm clearly detects both targets, located at 35 m and 70 m.

## 5. Performance Evaluation

### 5.1. Simulation Results

This section discusses the simulation results to confirm the performance improvement of the proposed algorithm. For all simulations, the center frequency $f_{0}$ is set to 24 GHz and the complex amplitude $a_{m}$ in Equation (Equation 5) was randomly and independently generated from a uniform distribution. In other words, the magnitude and angle of $a_{m}$ are in the ranges $0\leq |a_{m}|\leq 1$ and $0\leq ∡a_{m}\leq 2\pi$, respectively. The size of the FFT for range estimation $N_{R}$ was set to 256. As measure for performance evaluation, the root mean square error (RMSE) and missing rate of range detection are employed [15]. Total of $10^{5}$ simulations were performed to obtain the results of RMSE and missing rate. The RMSE is calculated with $\mathrm{RMSE}=\sqrt{\frac{1}{M\times 10^{5}}\sum_{i=1}^{10^{5}}\sum_{m=1}^{M}{\left(\theta_{m}-{\hat{\theta }}_{m}\right)}^{2}}$. Missing rate is the probability that the number of obtained peaks by peak detection is lower than the number of targets.

In Figure 10a,b, a slow-moving target ($v_{r,m}=5$ km/h) and a fast-moving target ($v_{r,m}=60$ km/h) are considered, respectively. For the slow-moving target, the RMSE of the proposed algorithm is lower than the RMSE of the previous algorithm for SNR ≤ 8 dB. In the region with SNR ≥ 10 dB, the RMSEs of the two algorithms are similar. Meanwhile, for the fast-moving target, shown in Figure 10b, the RMSE of the proposed algorithm is significantly lower than the RMSE of the previous algorithm over the entire SNR range.

To evaluate the improvement of the proposed algorithm compared to the previous algorithm, the missing rate is shown in Figure 11. In Figure 11a,b, a slow-moving target ($v_{r,m}=5$ km/h) and a fast-moving target ($v_{r,m}=60$ km/h) are considered, respectively. In both of the conditions, the slow- and fast-moving targets, the missing rate of the proposed algorithm was lower than the missing rate of the previous algorithm over the entire SNR region.

### 5.2. Complexity Comparison

This section evaluates the computational complexity of the previous algorithm and the proposed algorithm. In the case of the previous algorithm, the range and angle detections are performed regardless of the existence of a target. Hence, the required number of multiplications of the previous algorithm $C_{\mathrm{previous}}$ is as follows:(13)$$\begin{array}{c} C_{\mathrm{previous}}=\frac{N_{R}}{2}\log_{2}N_{R}\times K+N_{R}\times \frac{N_{A}}{2}\log_{2}N_{A}. \end{array}$$

On the other hand, the proposed algorithm performs angle detection only when it is determined that a target exists. Hence, the required number of multiplications of the proposed algorithm $C_{\mathrm{proposed}}$ is as follows:(14)$$\begin{array}{c} C_{\mathrm{proposed}}=\frac{N_{R}}{2}\log_{2}N_{R}+p_{T}\left(\left(K-1\right)\frac{N_{R}}{2}\log_{2}N_{R}+N_{R}\times \frac{N_{A}}{2}\log_{2}N_{A}\right), \end{array}$$
where $p_{T}$ is the probability that a target exists. Figure 12 shows the required number of multiplications according to the probability that the target exists. The target is not always present and, thus, the power efficiency of the proposed algorithm is always improved compared to the previous algorithm.

## 6. Experiments

To verify the effectiveness of the proposed algorithm in a practical environment, we performed experiments inside an anechoic chamber, located at DGIST in Korea. This section consists of two subsections. First, the experimental equipment is introduced, and then the experimental results are addressed.

### 6.1. Experimental Setup

We used an FMCW radar system at 24 GHz, which has two TX antennas and eight RX antennas, as shown in References [11,15]. Figure 13 shows a block diagram of the front-end module (FEM). The radio frequency (RF) module is composed of TX and RX sides, as shown in Figure 13. A microcontroller unit (MCU), frequency synthesizer with a phase-locked loop (PLL), and voltage-controlled oscillator (VCO) were included on the TX side, with a maximum bandwidth of 2 GHz. The MCU chip controls the frequency synthesizer with the PLL. Finally, the output of the VCO is connected to the two TX antennas through a power amplifier (PA).

The RX side includes the eight RX antennas, low-noise amplifiers (LNAs), high-pass filters (HPFs), a variable-gain amplifier (VGA), and low-pass filters (LPFs). The outputs of the LNAs are multiplied by the TX signals, and the outputs then pass through the HPFs with 150 kHz of bandpass frequency. The outputs of the HPFs are amplified by the amplifiers, and then the eight beat signals from different channels were obtained and the amplified signals were passed through the LPFs with 1.7 MHz of bandpass frequency. The noise of the RX was 8.01 dB and the RX antenna gain was 10 dB. The RX antenna azimuth beamwidth was 99.6${}^{{}^{\circ}}$ and elevation beamwidth was 9.9${}^{{}^{\circ}}$ [11,15]. A more detailed specification of the TX and RX sides is described in References [11,15].

Figure 14 shows the FEM and back-end module (BEM) systems for the experiment. In Figure 14a, the digital signal processing (DSP) and a field programmable gate array (FPGA) operating at up to 1 GHz are included in the data-logging board. First, the analog signal is converted to digital data in up to eight channels, with a 20 MHz sampling rate through the ADC. Two external memories of the DSP, the 2 GB DDR2 SDRAMs, provide a total of 512 Mbytes of data storage space. When the external memory is filled, the data is transferred to the computer through the Ethernet.

Figure 15 shows a photograph of an experiment in the chamber. As can be seen in Figure 15, the experiment was performed inside an anechoic chamber to avoid the undesired echo effect. This chamber was designed for 8 to 110 GHz, and its size was 5 (W) × 10 (L) × 4 m (H). The targets move back and forth depending on the speed set by the user. The duration of the chirp (ramp) *T* was set to 400 μs, the bandwidth was set to 1 GHz, and the sampling frequency was set to 5 MHz. The number of chirps per one frame was set to 256 and the number of frames was set to 64. A 2048-point FFT was performed for range estimation, and a 256-point FFT was performed in the DOA estimation step.

### 6.2. Experiment Results

This section addresses the experimental results to confirm the improvement induced by the proposed algorithm. Figure 16 shows the subtraction between two beat signals, $d_{k}^{\left(i\right)}\left[n\right]$, and the range detection result by the FFT of the previous algorithm with $l_{1}=6$ and $l_{2}=29$, for a single-target condition. The velocity of the target was set to 5 km/h. As shown in Figure 16a, we do not observe a sinusoidal signal form, although $d_{k}^{\left(i\right)}\left[n\right]$ should be a sinusoidal form for a single target. In Figure 16b, we observe that the previous algorithm does not detect the target at the actual range. On the other hand, Figure 17 shows the results of the same experiment using the proposed algorithm. In Figure 17a, it can be seen that $d_{k}^{\left(i\right)}\left[n\right]$ by the proposed algorithm is approximately a sinusoidal form. Figure 17b also shows the proposed algorithm accurately detects the range of target.

Figure 18 shows the experimental results for the condition of multiple targets with different velocities. The velocities of the fast- and slow-moving targets were set to 14.4 and 5 km/h, respectively. In the case of the previous algorithm, the indices of two beat signals were set to $l_{1}=6$ and $l_{2}=33$. From Figure 18a, it can be seen that the previous algorithm detects only one target despite the presence of two targets. On the other hand, both of the targets were accurately detected in the proposed algorithm as shown in Figure 18b.

## 7. Conclusions

We proposed an FMCW surveillance radar algorithm that not only solves the blind-speed problem but also reduces complexity. We showed that the previous algorithm misses a moving target because the previous algorithm did not consider the velocity of the target. Through simulation results, we showed that the proposed algorithm not only distinguishes between stationary and moving targets, but it also solves the problem of missing a target at a certain velocity despite low complexity. Furthermore, we set up an experiment environment using the FMCW radar system to verify the effectiveness of the proposed algorithm in a real environment. The experiment results showed that the proposed algorithm achieves better performance compared to the previous algorithm in a real environment. This algorithm has the disadvantage that it cannot measure the velocity of the target. Currently, we are proceeding with further study on algorithms that detect the speed of the target without increasing the complexity.

## Figures and Tables

**Figure 1 sensors-19-00608-f001:**
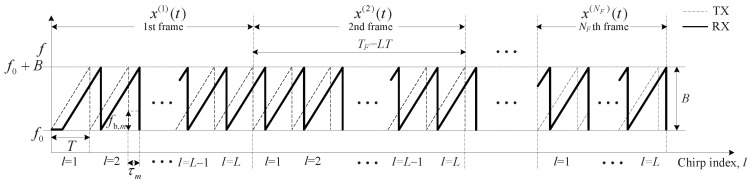
Structure of transmitted (TX) and received (RX) signals in frequency-modulated continuous wave (FMCW) radar; $f_{0}$ is carrier frequency, *B* is bandwidth, $x^{\left(i\right)}\left(t\right)$ is FMCW chirp signal at the *i*th frame, *T* is duration of one FMCW chirp signal, $T_{F}$ is duration of one frame, $N_{F}$ is the number of total frames and $\tau_{m}$ is the time delay of the *m*th target and $f_{b,m}$ is the beat frequency of the *m*th target.

**Figure 2 sensors-19-00608-f002:**
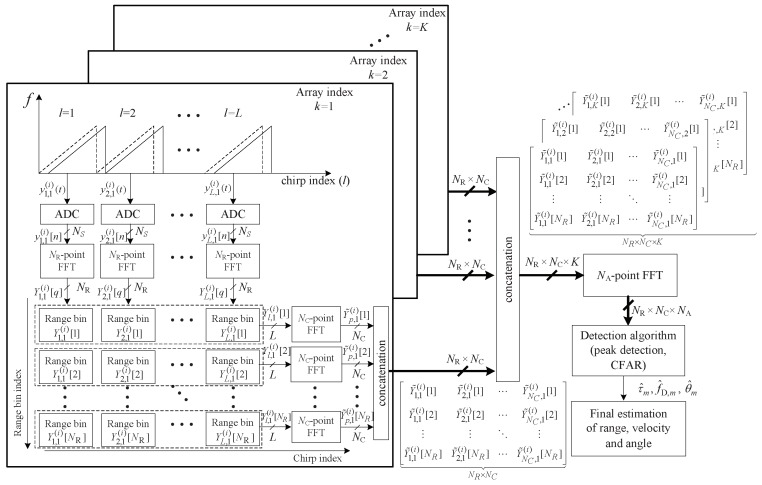
Structure of a three-dimensional (3D) fast Fourier transform (FFT) in FMCW radar.

**Figure 3 sensors-19-00608-f003:**
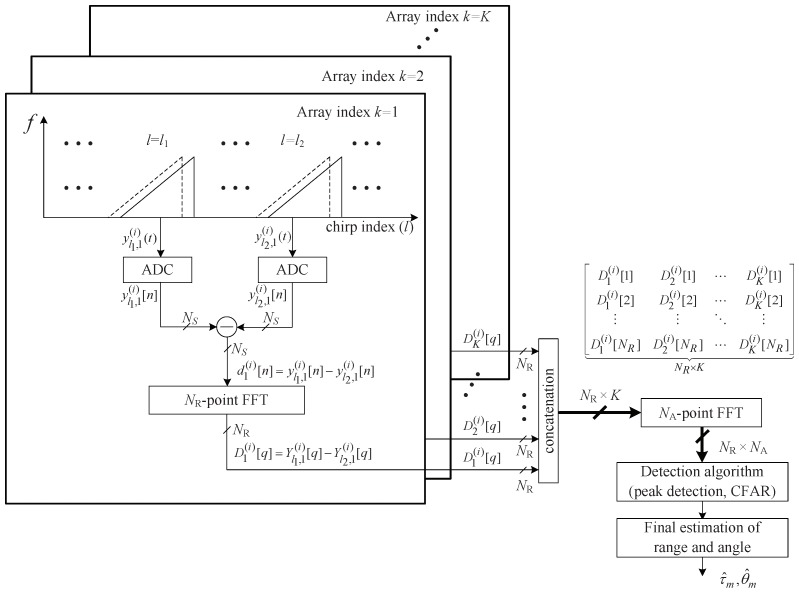
Structure of a low-complexity surveillance FMCW radar algorithm using dual chirps.

**Figure 4 sensors-19-00608-f004:**
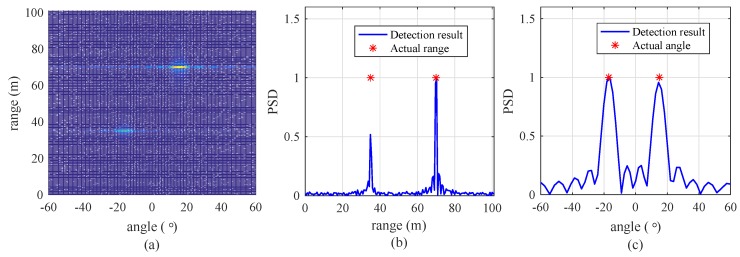
Snapshot of detection results with a SNR of 20 dB, $N_{R}$ = 512, and $N_{A}$ = 64. (**a**) Detection result using a 2D FFT, (**b**) result of range detection by a FFT, and (**c**) result of angle detection by a FFT.

**Figure 5 sensors-19-00608-f005:**
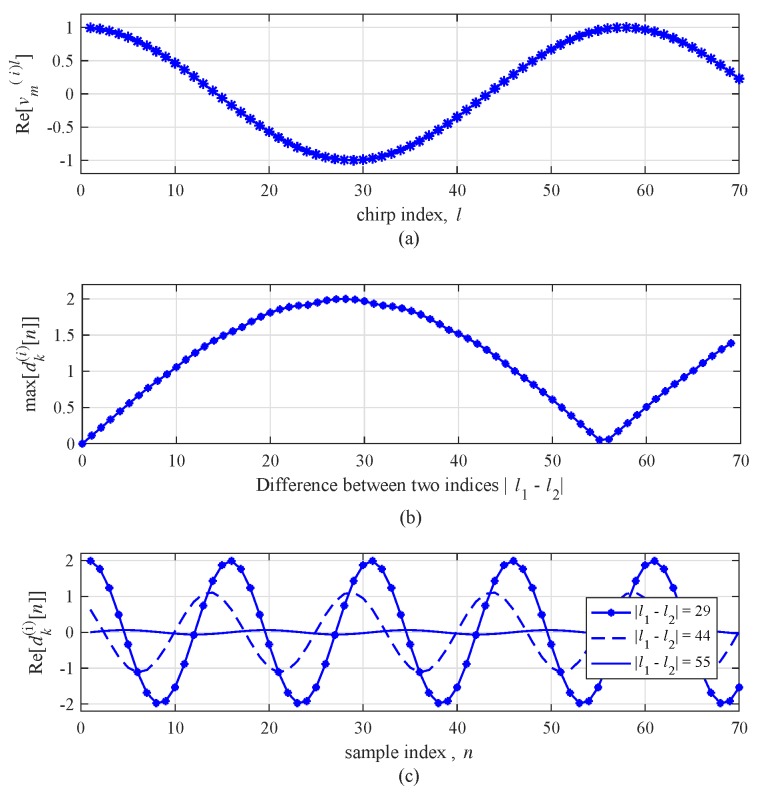
Real part of $v_{m}^{(i)l}$ without noise for a target with $v_{r,m}=$ 5 km/h: (**a**) amplitude with respect to the index *l*, (**b**) max value of $d_{k}^{\left(i\right)}\left[n\right]$ with respect to $|l_{1}-l_{2}|$, and (**c**) $d_{k}^{\left(i\right)}\left[n\right]$ for fixed $|l_{1}-l_{2}|$ with respect to *n*.

**Figure 6 sensors-19-00608-f006:**
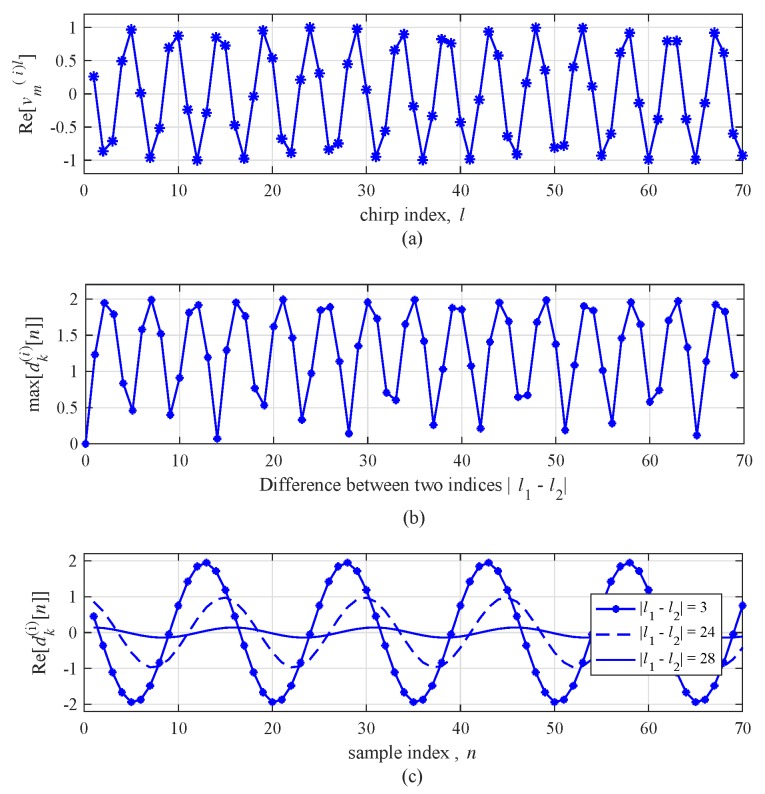
Real part of $v_{m}^{(i)l}$ without noise for a target with $v_{r,m}=$ 60 km/h: (**a**) amplitude with respect to *l*, (**b**) max value of $d_{k}^{\left(i\right)}\left[n\right]$ with respect to $|l_{1}-l_{2}|$, and (**c**) $d_{k}^{\left(i\right)}\left[n\right]$ for fixed $|l_{1}-l_{2}|$ with respect to *n*.

**Figure 7 sensors-19-00608-f007:**
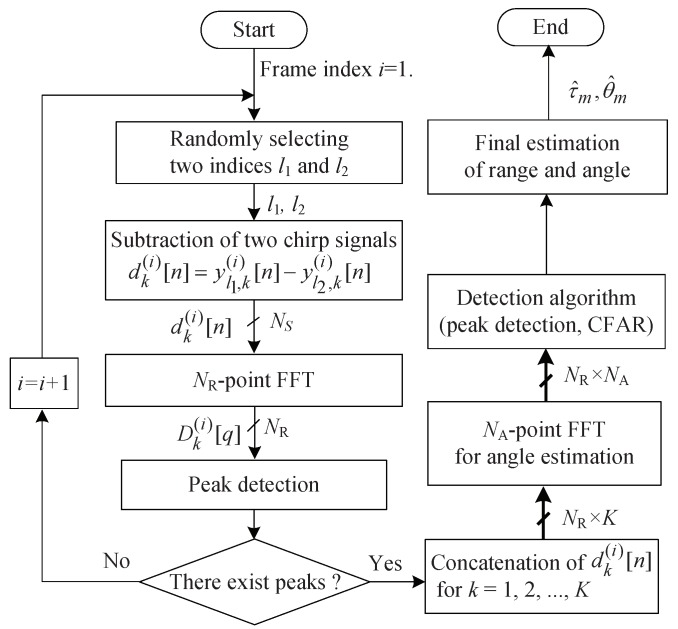
Structure of the proposed algorithm.

**Figure 8 sensors-19-00608-f008:**
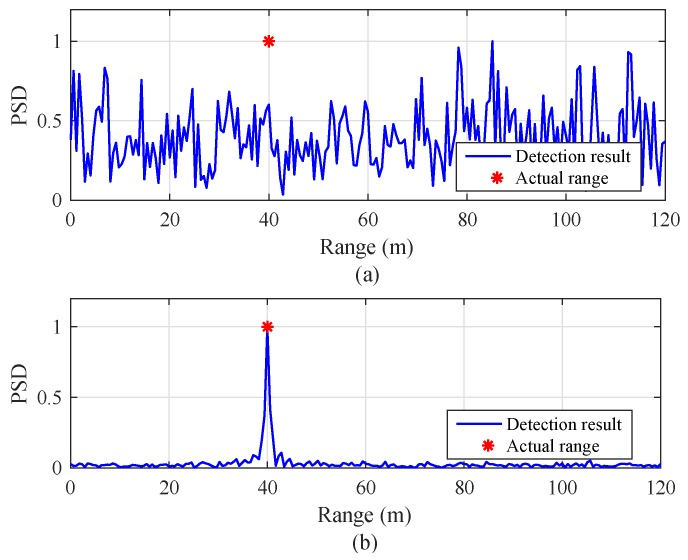
Snapshot of range detection results with SNR = 5 dB, $N_{R}$ = 256, and $v_{r,m}^{\left(i\right)}=60$ km/h: (**a**) the previous algorithm [8] ($l_{1}=1$, and $l_{2}=25$), and (**b**) the proposed algorithm.

**Figure 9 sensors-19-00608-f009:**
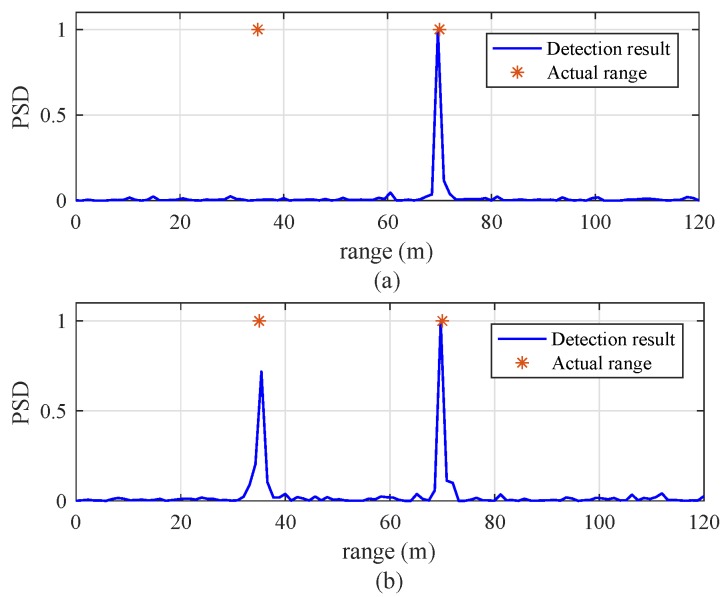
Snapshot of range detection results for two targets with $v_{r,1}^{\left(i\right)}=3$ km/h, $v_{r,2}^{\left(i\right)}=10$ km/h, SNR = 5 dB, and $N_{R}$ = 256: (**a**) previous algorithm [8] ($l_{1}=1$, and $l_{2}=54$), (**b**) proposed algorithm.

**Figure 10 sensors-19-00608-f010:**
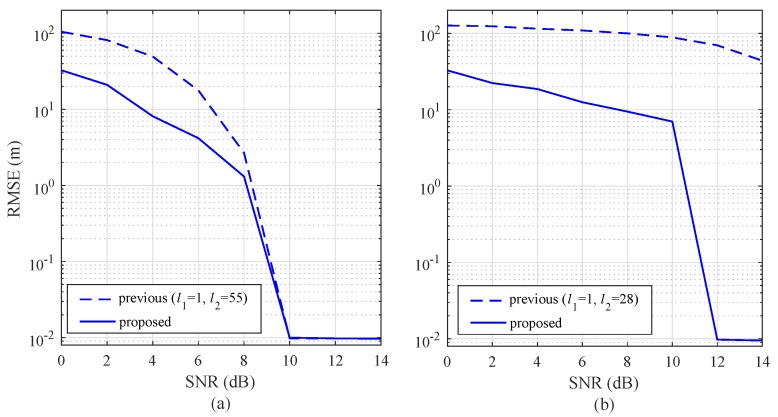
Root mean square error (RMSE) with respect to SNR for $N_{R}$ = 256: (**a**) $v_{r,m}=$ 5 km/h, and (**b**) $v_{r,m}=$ 60 km/h.

**Figure 11 sensors-19-00608-f011:**
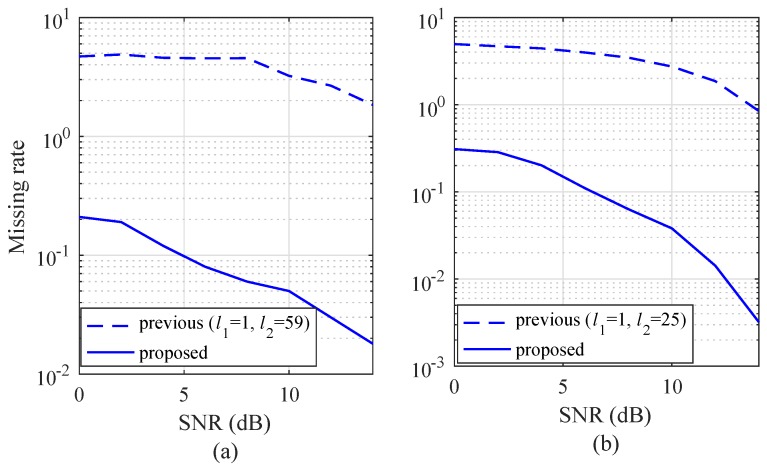
Missing rate as a function of SNR with $N_{R}$ = 256: (**a**) $v_{r,m}=$ 5 km/h, and (**b**) $v_{r,m}=$ 60 Km/h.

**Figure 12 sensors-19-00608-f012:**
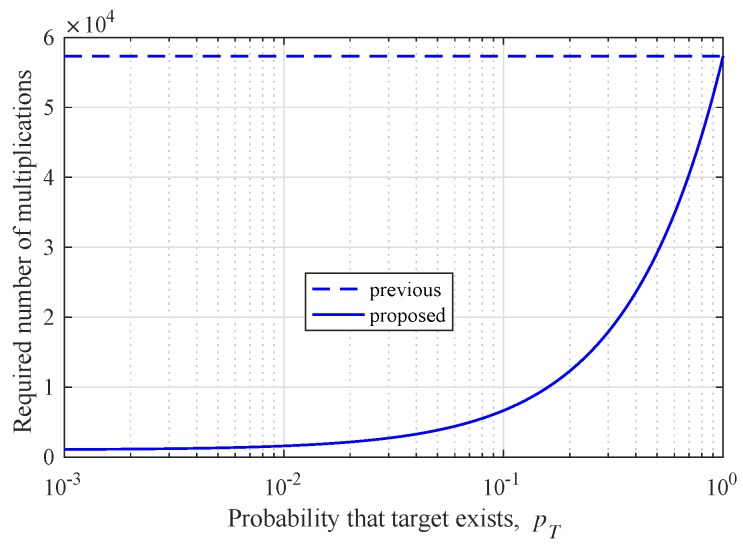
Required number of multiplications according to the probability that the target exists.

**Figure 13 sensors-19-00608-f013:**
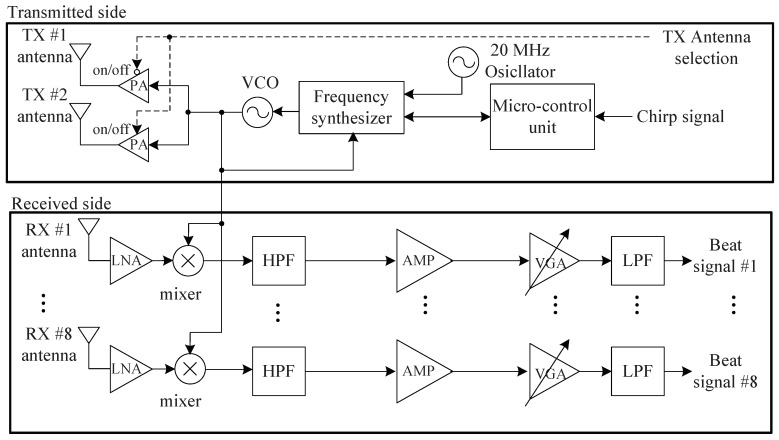
Block diagram of the 24 GHz radar RF module.

**Figure 14 sensors-19-00608-f014:**
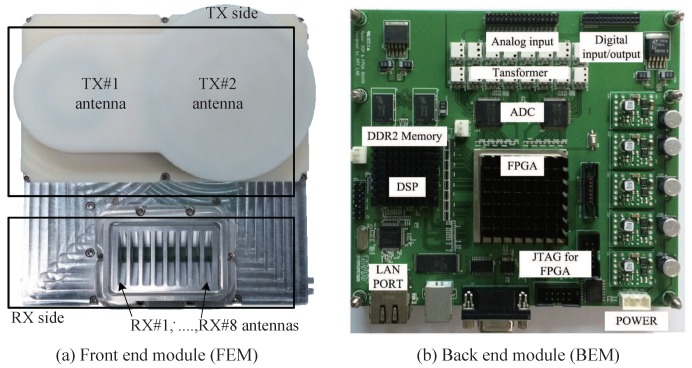
Front-end module and back-end module for the experiment.

**Figure 15 sensors-19-00608-f015:**
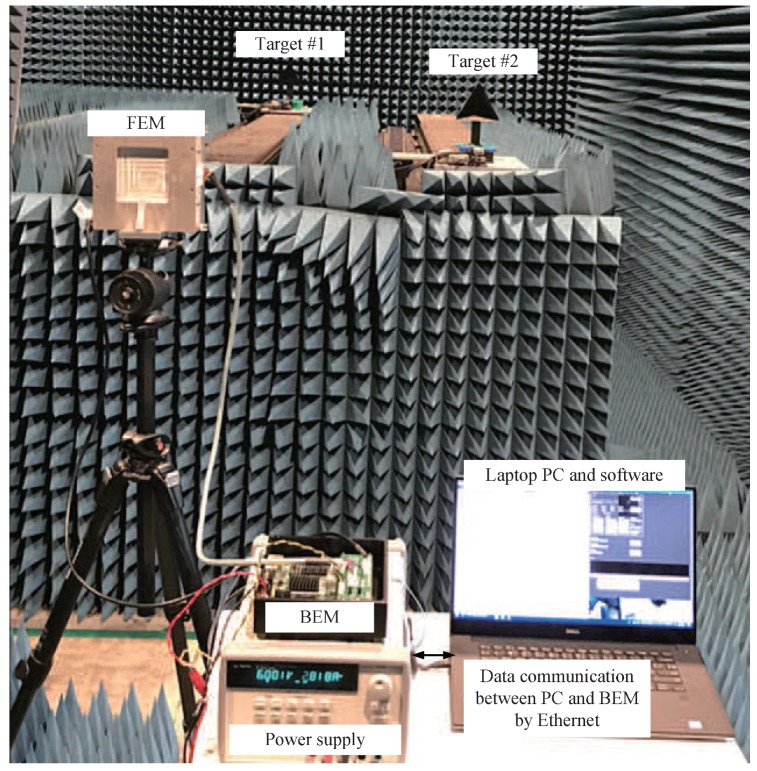
A photographs of the experiment setup in anechoic chamber.

**Figure 16 sensors-19-00608-f016:**
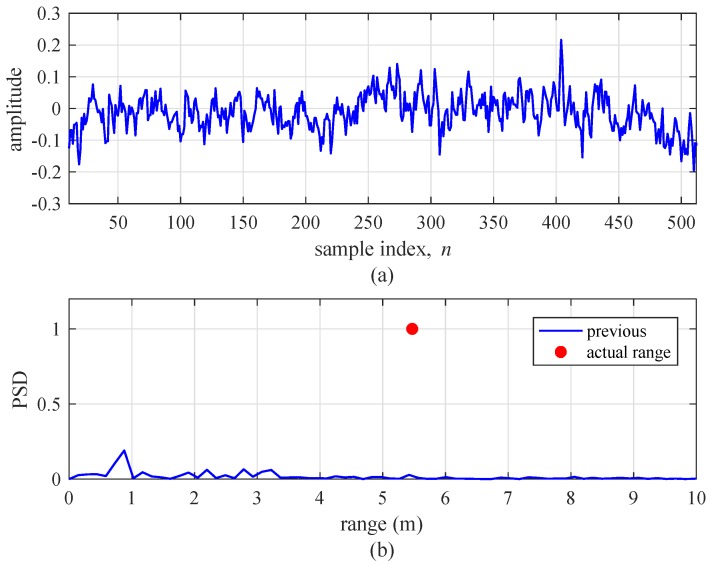
Experimental results of the previous algorithm ($l_{1}=6$ and $l_{2}=29$) for a single target. (**a**) Beat signal and (**b**) detection results using FFT.

**Figure 17 sensors-19-00608-f017:**
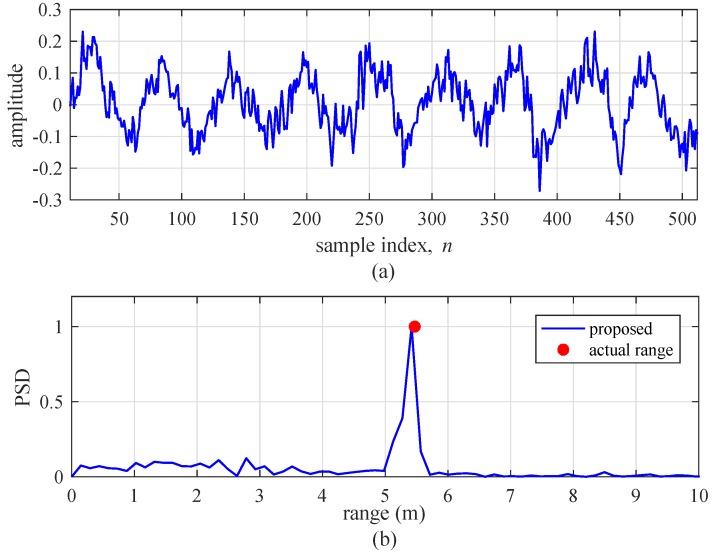
Experimental results of the proposed algorithm for a single target. (**a**) Beat signal, and (**b**) detection results using FFT.

**Figure 18 sensors-19-00608-f018:**
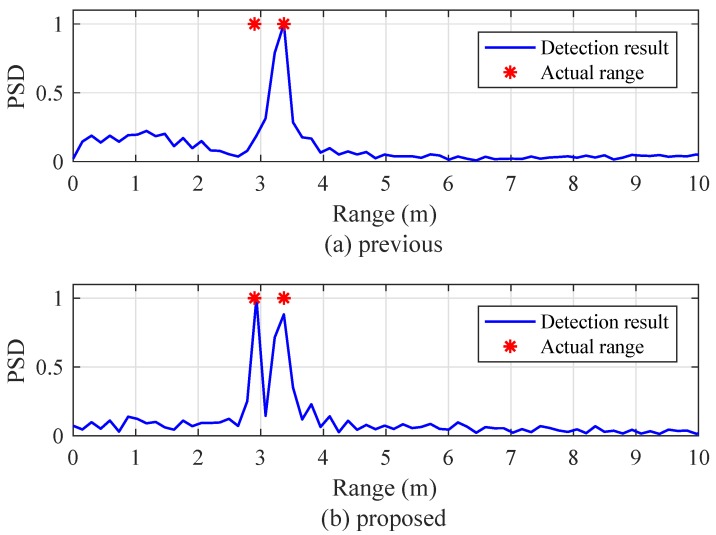
Experimental results for multiple targets. (**a**) Previous algorithm ($l_{1}=6$ and $l_{2}=33$), and (**b**) proposed algorithm.

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
