# Peer review of "A Low-Complexity FMCW Surveillance Radar Algorithm Using Two Random Beat Signals"

_sensors, 2019, doi:10.3390/s19030608_

Reviewer 1 Report

The paper presents a very simple algorithm to overcome the blind speed problem of MTI algorithm in frequency-modulated continuous wave radars. Despite its simplicity it seams to present an interesting performance. The document is not well written, it presents a considerable number of mistakes. I suggest authors to revise all the document in order to the paper be accepted for publication.

Author Response

We would like to acknowledge the thorough review and constructive comments made by the anonymous reviewers. We also thank the editor for great efforts coordinating the review and for taking time to read the manuscript. We have revised the manuscript by incorporating all reviewers' comments.

Comment 1: The paper presents a very simple algorithm to overcome the blind speed problem of MTI algorithm in frequency-modulated continuous wave radars. Despite its simplicity it seams to present an interesting performance. The document is not well written, it presents a considerable number of mistakes. I suggest authors to revise all the document in order to the paper be accepted for publication.

Response: Accepted. According to this comment, we have reviewed the entire document and newly revised. More specifically,

1. We tried to make the confusing sentences and expressions as clear as possible.

2. We tried to clarify the meaning of equations by dividing a confused equation into two equations and changing the forms.

3. We checked many typos and newly edited them in the entire document.

We have marked the parts modified in red color.  Please refer to the red parts in the revised manuscript. 

Thank you !

Reviewer 2 Report

This paper aims to solve the blind speed problem of moving target indicator (MTI) algorithm, while reducing the redundant complexity. Below are my suggestions or concerns:

1. There is a lack of description or brief discussion of lots of references. There are 14 references, but the authors referenced twelve of them ([1–11,13]) in the very beginning of the paper, and then never discussed some of them.

2. I would recommend that the authors can use subsections to further divide some of the sections, to make the organization clearer for readers to follow.

3. Fig. 4 shows a successful case of the proposed algorithm. However, since it is only one implementation of the algorithm, I think it would be better if there can be some results on the average effectiveness of the algorithm in a good number of implementations under random conditions.

4. How should |l_1-l_2| be choosed? Some further discussions would help the reader.

Author Response

Reply to Reviewer2’s Comments

sensors-425918

A Low-Complexity FMCW Surveillance Radar Algorithm Using Two Random Beat Signals

Bong-seok Kim, Youngseok Jin, Sangdong Kim and Jonghun Lee

We would like to acknowledge the thorough review and constructive comments made by the anonymous reviewers. We also thank the editor for great efforts coordinating the review and for taking time to read the manuscript. We have revised the manuscript by incorporating all reviewers' comments.

Comment 1: There is a lack of description or brief discussion of lots of references. There are 14 references, but the authors referenced twelve of them ([1–11,13]) in the very beginning of the paper, and then never discussed some of them.

Response: Accepted. According to this comment, we have tried to increase the descriptions or discussion of lots of references. To this end, we newly added new paragraphs and changed the structure of introduction. Please refer to introduction in the revised manuscript.

Comment 2: I would recommend that the authors can use subsections to further divide some of the sections, to make the organization clearer for readers to follow.

Response: Accepted. According to this comment, we used subsections to further divide some of the sections, to make the organization clearer for readers to follow. For example, we divided Section 2 into two subsections. Please refer to the blue marked titles of subsections in page 2 and page 4.  In addition, we divided Section 3 into two subsections. Please refer to the blue marked titles of subsections in page 5 and page 7.

Comment 3: Fig. 4 shows a successful case of the proposed algorithm. However, since it is only one implementation of the algorithm, I think it would be better if there can be some results on the average effectiveness of the algorithm in a good number of implementations under random conditions.

Response: Accepted. According to this comment, we newly mentioned root mean square error (RMSE) and missing rate in order to show the results on the average effectiveness of the algorithm in a good number of implementations under random conditions. 105 simulations for RMSE and missing rate were performed, respectively. Please refer to red parts and Fig. 10 and Fig. 11, in the revised manuscript.

Moreover, we newly included following sentence; Fig. 4 is the snapshot of detection results and the results on the average effectiveness of algorithms under random condition will be shown in Section 5. Please refer to the blue fonts at the last sentence in the page 6 in the revised manuscript.

Comment 4: How should |l1l2| be chosen? Some further discussions would help the reader.

Response: Accepted. According to this comment, we newly included further discussions of the choice of  |l1- l2| in order to help the reader in the revised manuscript. We have revised the explanations for the choice of  |l1- l2| in conditions that do not affect performance. Please refer to the red parts from the 2nd paragraph in page 8 to the 3rd paragraph in page 9.

Thank you !

Reviewer 3 Report

The article is well written clearly in the presentation. The experimental measurements are obtained in the laboratory, ie in an environment with well-defined characteristics. In a future work it would be desirable to test the algorithm in real operational condions of the FMCW radar.

Text editing corrections

line 6: insert the dot after "velocity"

line 32: errata "system [12]" corrige "system [12]"

Author Response

Reply to Reviewer3’s Comments

sensors-425918

A Low-Complexity FMCW Surveillance Radar Algorithm Using Two Random Beat Signals

Bong-seok Kim, Youngseok Jin, Sangdong Kim and Jonghun Lee

We would like to acknowledge the thorough review and constructive comments made by the anonymous reviewers. We also thank the editor for great efforts coordinating the review and for taking time to read the manuscript. We have revised the manuscript by incorporating all reviewers' comments.

Comment 1: Text editing corrections

line 6: insert the dot after "velocity" and  line 32: From "system[3]" to "system [3]" in the submitted manuscript.

Response: Accepted. According to this comment, we newly revised the manuscript.  Please refer to blue marked parts at the 6th line in Abstract and the 3th paragraph in page 2, respectively, in the revised manuscript.

Thank you !
